# Inhibitory Effects of the Addition of KNO_3_ on Volatile Sulfur Compound Emissions during Sewage Sludge Composting

**DOI:** 10.3390/bioengineering9060258

**Published:** 2022-06-17

**Authors:** Guodi Zheng, Yuan Liu, Yongjie Li, Junwan Liu, Junxing Yang

**Affiliations:** 1Center for Environmental Remediation, Institute of Geographic Sciences and Natural Resources Research, Chinese Academy of Sciences, Beijing 100101, China; liuya.20s@igsnrr.ac.cn (Y.L.); 20049803@chnenergy.com.cn (J.L.); yangajx@igsnrr.ac.cn (J.Y.); 2College of Resources and Environment, University of Chinese Academy of Sciences, Beijing 100049, China; 3CCCC Ecologital Protection Investment Company, Beijing 100013, China; leeyjie@126.com; 4Guoneng Longyuan Environmental Co., Ltd., Beijing 100039, China

**Keywords:** arylsulfatase, composting, hydrogen sulfide, odor, potassium nitrate

## Abstract

Odor released from the sewage sludge composting process often has a negative impact on the sewage sludge treatment facility and becomes a hindrance to promoting compost technology. This study investigated the effect of adding KNO_3_ on the emissions of volatile sulfur compounds, such as hydrogen sulfide (H_2_S), dimethyl sulfide (DMS), and carbon disulfide (CS_2_), during sewage sludge composting and on the physicochemical properties of compost products, such as arylsulfatase activity, available sulfur, total sulfur, moisture content, and germination index. The results showed that the addition of KNO_3_ could inhibit the emissions of volatile sulfur compounds during composting. KNO_3_ can also increase the heating rate and peak temperature of the compost pile and reduce the available sulfur loss. The addition of 4% and 8% KNO_3_ had the best effect on H_2_S emissions, and it reduced the emissions of H_2_S during composting by 19.5% and 20.0%, respectively. The addition of 4% KNO_3_ had the best effect on DMS and CS_2_ emissions, and it reduced the emissions of DMS and CS_2_ by 75.8% and 63.0%, respectively. Furthermore, adding 4% KNO_3_ had the best effect from the perspective of improving the germination index of the compost.

## 1. Introduction

In the process of sewage sludge composting, the uneven supply of oxygen inside the compost leads to local anaerobic pockets Han et al. [1]. Anaerobic conditions are more prone to producing a large number of odor gases [2], which brings adverse effects to the environment [3,4,5]. According to their chemical composition, odorous gases can be divided mainly into sulfides and nitrides [6]. In the past, studies have usually focused on controlling the release of nitrides, such as NH_3_ and N_2_O, to increase the nitrogen content of compost products [7,8,9]. However, volatile sulfur compounds (VSCs), especially hydrogen sulfide (H_2_S), dimethyl sulfide (DMS), carbon disulfide (CS_2_), methyl mercaptan (CH₃SH), and dimethyl disulfide (DMDS), are the main cause of odor pollution during sewage sludge composting due to their exceptionally low olfactory threshold and extremely high emissions [2,10,11]. Therefore, it is essential to achieve the control of VSCs. Sulfide biotransformation is mainly related to the oxidation of low-valent sulfur through sulfate-oxidizing bacteria (SOB) and the reduction of sulfate through sulfate-reducing bacteria (SRB) [12,13]. The activity of SOB and SRB is closely related to the pH and oxidation–reduction potential of raw compost materials [14,15]. Therefore, this activity can be changed by adding biological agents [16] and chemical reagents [17,18] to reduce the release of VSCs. For example, Zhang et al. [19] found that the addition of FeCl_3_ (10.0% in moles of the initial total nitrogen) to compost results in a 52.0% reduction in H_2_S emissions. Chen et al. [20] found that the addition of Fe_2_O_3_ (7.5% in moles of the initial total nitrogen) to the composting process reduced the emissions of H_2_S, DMDS, and DMS by 38.8%, 73.6%, and 42.6%, respectively.

There have been many studies on the inhibition of SRB activity through KNO_3_ to reduce the emission of H_2_S in wastewater pipes [21,22]. However, these studies mainly focused on water treatment. During the composting of sewage sludge, the solid nature may adversely affect the KNO_3_-inhibiting activity of SRB. Therefore, it is still necessary to study the practical effect of KNO_3_ addition on VSC control during sewage sludge composting. Moreover, adding KNO_3_ to compost can solve the problem of nitrogen and potassium deficiency when compost products are used as an organic fertilizer [23]. Furthermore, the increased costs of using additives can be compensated for during the application phase of the product.

Based on the relevant principles and research progress, we explored the effects of arylsulfatase in the conversion of sulfur and investigated the effects of the addition of KNO_3_ on the compost nutrient composition, dewatering effect, and product maturity by analyzing the effects of the addition of KNO_3_ on the emission of H_2_S, DMS, and CS_2_ during sewage sludge composting. The appropriate amount of KNO_3_ to add to the composting process was determined according to the inhibitory effects on the emission of H_2_S, DMS, and CS_2_. The findings of this study provide a theoretical basis for the control of odor pollution and the improvement of the quality of products by KNO_3_ in sewage sludge composting.

## 2. Materials and Methods

### 2.1. Composting Materials

The sewage sludge was collected from a municipal wastewater treatment plant in Zhengzhou, China, and it was dewatered to a moisture content of approximately 80%. The compost conditioner was sawdust (moisture content of approximately 10%). The mixture of sewage sludge and sawdust was 1:0.3 (*w*/*w*).

### 2.2. Composting Method and Device

The composting experiments were conducted in automatically controlled containers. The container consisted of a composting pot, a fan, and an online temperature monitoring system, as shown in Figure 1. The composting pot was made of PVC, covered with mineral wool insulation on the outside, and had an effective volume of 340 L (height, 1.2 m; inner diameter, 0.60 m). There are three temperature probes, each measuring the temperature of the pile at different depths in real-time. Ventilation was forced intermittent ventilation with 20 min intervals per 1 min of running. The ventilation rate was set at 3 L/min. The sewage sludge, sawdust and different amounts of additives (4%, 8%, and 12% KNO_3_, based on the wet weight of the sludge) were mixed thoroughly with a tool such as a shovel in an open area and then added to the composting container. The pile without the addition of KNO_3_ was taken as a control batch. The complete composting period was 15 days. On days 0–5, 7, 9, 11, 13 and 15 of the composting process, the upper, middle and lower parts of the compost pile were sampled using a sampler. At the same time, the released gas was collected at the gas outlet of the composting unit using a sampling bag. Collection times were 08:00 and 18:00, with at least one ventilation cycle per collection.

### 2.3. Analytical Methods

The composition and concentration of volatile organic sulfur compounds were measured using a portable gas chromatography-mass spectrometer (GC-MS, Infinicon, East Syracuse, NY, USA). The instrument was calibrated with sulfide mixed standard gas (6 substances). However, only DMDS and CS_2_ were continuously detectable in the experiments, so only the results for these two compounds would be discussed. The sampling bag was connected internally to the portable GC-MS sampling handle, and the analytical method was run. Gas samples were first concentrated using a TRI-BED concentrator (containing activated carbon, silica gel, Tanex) and analyzed in 100 mL injection volumes using high purity nitrogen as the carrier gas [24]. The H_2_S concentration was detected online using a portable H_2_S detector (BW08-III; range, 0–200 ppm; accuracy, 0.1 ppm).

The moisture content was determined by the oven drying method. The arylsulfatase activity was measured as described in the study of Alef and Nannipieri [25] in 0.5 M acetate buffer pH 5.8 using p-nitrophenyl sulfate as a substrate for 1 h, at 37 °C. The total sulfur was measured using a fractional elemental analyzer (FLASH 2000, Thermo Fisher Scientific, MA, USA). The available S was determined by reference to the turbidimetric method used in the study by Bao et al. [26]. Available S was extracted by shaking a 2.000 g compost sample with 50 mL Ca(H_2_PO_4_)_2_–HoAc solution for 1 h, and then filtered through a slow filter paper. The following specific instructions were followed: Pipette 10 mL of filtrate into a 100 mL conical flask, heat on an electric hot plate, add 30% H_2_O_2_ dropwise 3–5 times (to oxidize the organic matter), wait for the organic matter to decompose completely, then continue boiling and evaporate the bottom of the flask thoroughly to remove the excess H_2_O_2_. Add 1 mL of 1:4 HCl to obtain a solution. Transfer the solution together into a 25 mL volumetric flask and wash the solution in the conical flask several times. Add 2 mL of 2.5 g/L aqueous Gum Arabic solution and fix with ultrapure water. Transfer to a 50 mL beaker, add 1.0 g BaCl_2_-2H_2_O and stir with an electromagnetic stirrer for 1 min. Allow to stand for 20 min, then immediately remove the solution from the cuvette and turbidify at 440 nm. 

The seed germination index (GI) was measured as follows. Firstly, we mixed 5 g of the sample with 50 mL of distilled water, shaken for 2 h at ambient temperature and then centrifuged and filtered to obtain the extraction solution. We took 15 mL of the above extraction solution and placed it in a 12 cm diameter filter paper Petri dish. Then, we placed 20 cabbage seeds evenly in the Petri dishes. After placement, the Petri dishes were placed in an incubator and incubated for 48 h at 25 °C protected from light, and then the germination rate and root length of the seeds were measured. A blank control group was set up, and 15 mL of distilled water was added to a 12 cm diameter Petri dish with filter paper and incubated and measured in the above manner. The GI was calculated using the formula:GI (%) = [Seed germination of treatment (%) × Root length of treatment]/[Seed germination of control (%) × Root length of control (%)].

### 2.4. Statistical Analysis

All measurements were carried out in triplicate. The average value and standard deviation of the data were calculated using Microsoft Excel 2016. Correlation analysis was performed using SPSS v.21. Figures were created using OriginLab 2018.

## 3. Results and Discussions

### 3.1. Effect of KNO_3_ on the Temperature of the Composting Process

Temperature directly affects the activity and microbial community structure in the pile, which, in turn, affects the decomposition rate and the decay process of organic matter [27]. The composting process must reach a specific temperature and last for a certain amount of time to kill pathogenic bacteria. The temperature changes inside the piles with the addition of different amounts of KNO_3_ are shown in Figure 2. The control batch and treatments with the addition of 4%, 8%, and 12% KNO_3_ entered the thermophilic phase (above 50 °C) on the second, first, second, and third days and lasted for 3, 14, 4, and 5 days, respectively, which met the requirements for the duration of the high-temperature phase of sludge composting in the Standard for Sludge Stabilization Treatment of Municipal Wastewater Treatment Plant (CJ/T 510-2017). In contrast, the piles with added KNO_3_ all showed an increase in peak temperature and an extension of the high-temperature phase compared to the control batch. A possible reason is that K is an essential nutrient for microorganisms that can act as an activator of enzymes and a stabilizer of protein synthesis [28], thus increasing the microbial activity in the pile and promoting heat production. Additionally, the addition of KNO_3_ can change the composting environment and provide nutrients for microbial metabolism [29,30], which could be another reason. Under the treatment of the addition of 4% KNO_3_, the pile heated up the fastest and the peak temperature was the highest, which in part indicates that this amount had the best effect on the microbial metabolism. Compared to 4% KNO_3_, the treatments with the addition of 8% and 12% KNO_3_ showed slower warming of the pile and a decrease in peak temperature, indicating that the effect of this dose on the microbial metabolism was not as favorable as that of 4%. It is possible that different doses of KNO_3_ resulted in significantly different microbial structures and activity in the pile, which is consistent with the effect of KNO_3_ addition on the microbial community structure in chicken manure compost [29]. In general, the addition of KNO_3_ can indeed increase the heating rate and peak temperature of the pile, and the addition of 4% has the best effect.

### 3.2. Effect of KNO_3_ on H_2_S Emissions during the Composting Process

H_2_S is one of the main causes of malodorous compound emissions from sewage sludge composting because of its high emissions and low olfactory threshold [31]. The dynamics of H_2_S emissions under different KNO_3_ additions are shown in Figure 3a. The H_2_S emissions show a similar trend at different doses of KNO_3_ addition, that is, peak concentrations are reached at the start of composting and H_2_S emissions almost cease after 5 days. Furthermore, this trend is consistent with the studies of Gao et al. [32] and Xu et al. [33]. In the control batch and treatments comprising the addition of 4%, 8%, and 12% KNO_3_, the peak time of H_2_S emissions occurred on the second, second, second, and third days with peak concentrations of 49.7, 46.2, 28.4, and 18.2 mg·m^−3^, respectively. Compared to the control batch, the peak concentrations of H_2_S released from the piles with KNO_3_ additions of 4%, 8%, and 12% decreased by 7.0%, 42.9%, and 63.4%, respectively. The results showed that, as with the drainage network, KNO_3_ did reduce the production of H_2_S in the composting environment. The cumulative amounts of H_2_S in the control batch and treatments comprising the addition of 4%, 8%, and 12% KNO_3_ were 23.6, 19.0, 18.9, and 24.5 mg (Figure 3d). Compared to the cumulative amount of H_2_S from the pile of the control treatment, the cumulative amounts of H_2_S with the addition of 4% and 8% decreased by 19.5% and 20.0%, whereas with 12%, it increased by 3.5%.

In the composting process, H_2_S is produced mainly from the microbial reduction of sulfate and decomposition of VSC substrates under anaerobic conditions. Under anaerobic conditions, due to uneven oxygenation, SRB can reduce SO_4_^2−^ to S^2−^ [34,35]. The results showed that the addition of KNO_3_ in the sewage sludge composting process could reduce the release of H_2_S by approximately 20%. The first reason for the reduced release of H_2_S may be that the addition of KNO_3_ increases the electron acceptor and raises the oxidation–reduction potential, thereby reducing the production of H_2_S [36,37]. The second reason may be that KNO_3_ significantly inhibited the reproductive rate and sulfate-reducing activity of SRB [38]. The third reason may be that KNO_3_ can diffuse from the surface to the interior of the sludge particles and react as an electron acceptor in the role of SOB to produce S^0^ and SO_4_^2−^ [39]. The fourth reason may be that NO_3_^−^ can also directly oxidize S^2−^ to SO_4_^2−^ by chemical action (8H^+^ + 5S^2−^ + 8NO_3_^−^ → 5SO_4_^2−^ + 4N_2_ + 4H_2_O). It is worth pointing out that KNO_3_ seems to increase the release of H_2_S in the later stage of composting, which may be further evidence that KNO_3_ can influence the metabolic activities associated with H_2_S. According to the results of the above study, the addition of 4% and 8% KNO_3_ has the best inhibitory effect on the release of H_2_S.

### 3.3. Effect of KNO_3_ on DMS and CS_2_ Emissions during the Composting Process

DMS and CS_2_ are also typical malodorous gases [31,40], which can irritate the human respiratory tract, affect the physiological functions of the liver, kidneys, and cardiovascular system, and cause memory decline [41]. The dynamics of DMS and CS_2_ emitted from piles with different KNO_3_ additions are shown in Figure 3b,d. In the composting process, DMS and CS_2_ had similar trends, with their emissions occurring at the beginning of composting and later stopping in the middle of composting. In the control batch and treatments comprising the addition of 4%, 8%, and 12% KNO_3_, the peak time of DMS emissions occurred on the third, second, second, and third days, with peak concentrations of 586.3, 115.2, 99.2, and 254.7 μg·m^−3^, respectively. Compared to the control batch, the peak concentrations of DMS released from the piles with KNO_3_ additions of 4%, 8%, and 12% decreased by 80.4%, 83.1%, and 56.6%, respectively. In the control batch and treatments comprising the addition of 4%, 8%, and 12% KNO_3_, the peak time of CS_2_ emissions occurred on the third, first, first, and third days with peak concentrations of 1752.8, 456.3, 604.9, and 971.3 μg·m^−3^, respectively. Compared to the control batch, the peak concentrations of DMS released from the piles with KNO_3_ additions of 4%, 8%, and 12% decreased by 74.0%, 65.5%, and 44.6%, respectively.

The cumulative amounts of DMS in the control batch and treatments comprising the addition of 4%, 8%, and 12% KNO_3_ were 230.4, 55.7, 58.2, and 182.1 μg, respectively (Figure 3e). Compared to the cumulative amount of DMS from the pile of the control treatment, the cumulative amounts of DMS with the addition of 4%, 8%, and 12% decreased by 75.8%, 74.8%, and 21.0%, respectively. The cumulative amounts of CS_2_ in the control batch comprising treatments with the addition of 4%, 8%, and 12% KNO_3_ were 593.2, 219.4, 313.6, and 690.8 μg, respectively (Figure 3f). The cumulative amount of CS_2_ from the pile with the addition of 4% and 8% KNO_3_ decreased by 63.0% and 47.1%, respectively, whereas that from the pile with the addition of 12% KNO_3_ increased by 16.5%, compared to that from the control batch.

The pathways of metabolism for DMS are very complicated. According to the mechanism of sulfide conversion under anaerobic conditions proposed by Higgins et al. [42], DMS is mainly derived from the oxidation of methanethiol, which is derived from the decomposition of sulfate, cysteine, and methionine and the methylation of H_2_S. It has also been found that DMS did not show a decreasing trend with increasing biostability and oxygen content, inferring that its formation is mainly an abiotic process [13,30]. Based on the above studies, it can be inferred that a possible reason for KNO_3_ controlling the release of DMS in this study is the direct reduction in the production of H_2_S, a key precursor substance. The addition of 4% KNO_3_ inhibited the release of DMS more than it did the release of H_2_S; therefore, KNO_3_ may also reduce the release of DMS by inhibiting the conversion of H_2_S to DMS. There is a relative lack of studies on the conversion principle of CS_2_, and based on their similar results, it is inferred that the control principle of CS_2_ by KNO_3_ is the same as that of DMS. Combining the release concentrations and cumulative emissions of DMS and CS_2_, the appropriate (4%) addition of KNO_3_ has a remarkable inhibitory effect on the emissions of DMS and CS_2_.

### 3.4. Effect of KNO_3_ on the Activity of Arylsulfatase in the Composting Process

Arylsulfatase, an important enzyme involved in the sulfur metabolism during composting, releases SO_4_^2−^ by hydrolyzing the thioester bond (SO) in organic sulfur (ROSO_3_^−^ + H_2_O → ROH + SO_4_^2−^ + H^+^), and its activity is related to the concentrations of organic sulfur and SO_4_^2−^ [43,44]. The arylsulfatase activity showed a trend of increasing and then decreasing and was the highest in the early stage of sewage sludge composting (Figure 4). This trend is consistent with studies by Meena et al. [44] and Ma et al. [45]. The maximum arylsulfatase activity was 1.42, 1.45, 1.35, and 1.04 μg·g^−1^·h^−1^ in the control batch and the treatments comprising the addition of 4%, 8%, and 12% KNO_3_. During the whole composting process, there was no significant change in the activity of arylsulfatase in the pile with the addition of 4% KNO_3_ compared to the control batch, whereas the activity of arylsulfatase decreased significantly with the addition of 8% and 12% KNO_3_. There was a positive correlation (N = 30, *p* = 0.01, *R*^2^ = 0.64) between arylsulfatase activity and H_2_S release data for the control batch and treatments comprising the addition of 4% and 8% KNO_3_. There was a significant positive correlation (N = 10, *p* = 0.01, *R*^2^ = 0.82) between arylsulfatase activity and H_2_S release data for treatment with the addition of 8% KNO_3_. This result is consistent with the theory that arylsulfatase promotes the hydrolysis of organic sulfur and thus the release of H_2_S.

The mechanism of arylsulfatase activity regulation is currently unknown. In this study, it was hypothesized that the decrease in arylsulfatase activity was due to the addition of KNO_3_, which inhibited the sulfate reduction process dominated by SRB, followed by an increase in the SO_4_^2−^ concentration, which eventually led to a decrease in arylsulfatase activity. In short, KNO_3_ addition indirectly inhibits arylsulfatase activity during composting.

### 3.5. Effects of Adding KNO_3_ on the Total and Available Sulfur Contents

The total sulfur content showed a continuous decreasing trend throughout the composting process (Figure 5a). In the control batch and treatments comprising the addition of 4%, 8%, and 12% KNO_3_, the total sulfur content of the compost decreased from 3.28, 3.21, 3.24, and 3.23 g/kg on day 1 to 2.60, 2.75, 2.71, and 2.64 g/kg on day 15, respectively. They decreased by 20.7%, 14.3%, 16.4% and 18.3%. The results show that the total sulfur loss of the compost was reduced under the treatment of the addition of 4% KNO_3_, which may be related to the reduction of VSC odor emissions.

The available sulfur in compost can directly reflect the sulfur supply capacity of a plant, which contains water-soluble sulfur and a part of adsorbed sulfur [26]. The available sulfur content of the piles of all treatments showed a relatively smooth increasing trend (Figure 5b). In the control batch and treatments comprising the addition of 4%, 8%, and 12% KNO_3_, the available sulfur content of the compost increased from 1.60, 1.66, 1.75, and 1.92 g/kg on day 1 to 1.87, 2.01, 2.05, and 2.22 g/kg on day 15, respectively. They increased by 17.3%, 21.0%, 17.2%, and 16.2%. In summary, the addition of 4% KNO_3_ was the most effective in increasing the available sulfur content and reducing the total sulfur loss in the compost products.

### 3.6. Effect of Adding KNO_3_ on the Moisture Content and GI

The moisture content is one of the most important composting process parameters [46]. Furthermore, the moisture content affects the speed of the aerobic composting reaction, the degree of compost maturation, and the quality of compost products [47]. The moisture content of the control batch and treatments comprising the addition of 4%, 8%, and 12% KNO_3_ decreased from 63.9%, 62.2%, 61.4%, and 61.9% on day 1 to 57.2%, 56.8%, 55.9%, and 54.4% on day 15, respectively (Figure 6a). They decreased by 10.5%, 8.7%, 9.0%, and 12.1%. The relatively small amount of water removed in this study compared to the results of other studies, such as Chen et al. [48], may be due to the low ventilation frequency setting. In combination with the changes in the moisture content of the composting process and the moisture content of the compost products, it was found that the addition of KNO_3_ did not have an adverse effect on the dewatering of the compost.

GI is commonly used to assess the maturity and phytotoxicity of compost products [49]. Through the composting treatment, the GI of the compost products all increased (Figure 6b). In the control batch and in the treatments comprising the addition of 4%, 8%, and 12% KNO_3_, the GI of the compost products increased from 72.7%, 62.4%, 34.4%, and 20.2% on day 1 to 86.8%, 83.3%, 47.4%, and 40.7% on day 15, respectively. The results showed that the higher the addition of KNO_3_, the lower the GI of the compost products. Although the GI after the composting process were significantly improved under KNO_3_ additions of 8% and 12%, they still did not meet the requirement of GI of more than 60% in the Standard for Stabilization of Municipal Wastewater Treatment Plant Sludge (CJ/T 510-2017). A possible reason for this phenomenon is the presence of large amounts of KNO_3_ in the compost product, which generates ionic toxicity and osmotic stress that adversely affects the seed germination process [50]. In contrast to the above conclusions that the addition of KNO_3_ leads to an increase in microbial activity in the composting process, the negative effect of the addition of KNO_3_ on the seed germination process may be due to the fact that the process is more sensitive to such salt changes than the composting process. Chen et al. [51] found that the GI of *Platycodon grandiflorum* treated with high concentrations of KNO_3_ was significantly lower than that of the low-concentration treatment, which is consistent with the results of this study. In conclusion, the addition of 4% KNO_3_ did not adversely affect the decomposition of the compost product, but higher additions may.

## 4. Conclusions

The addition of 4% KNO_3_ in the sewage sludge composting process significantly reduced the emissions of VSCs and could increase the available and total sulfur of compost products. At this dose, there is no appreciable effect on the decomposition and dewatering of the compost. Moreover, the addition of 4% KNO_3_ in the composting process had the fastest warming rate, the longest high-temperature phase, and the best control of VSCs. The addition of KNO_3_ inhibited the activity of arylsulfatase during composting.

## Figures and Tables

**Figure 1 bioengineering-09-00258-f001:**
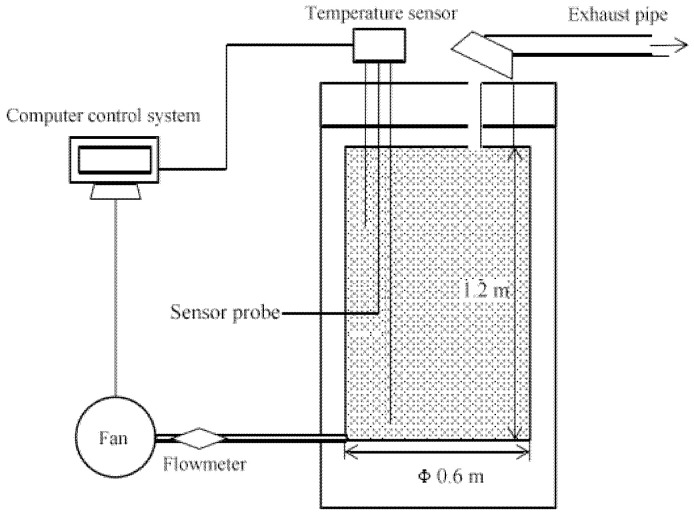
Schematic of aerobic composting automatic control container.

**Figure 2 bioengineering-09-00258-f002:**
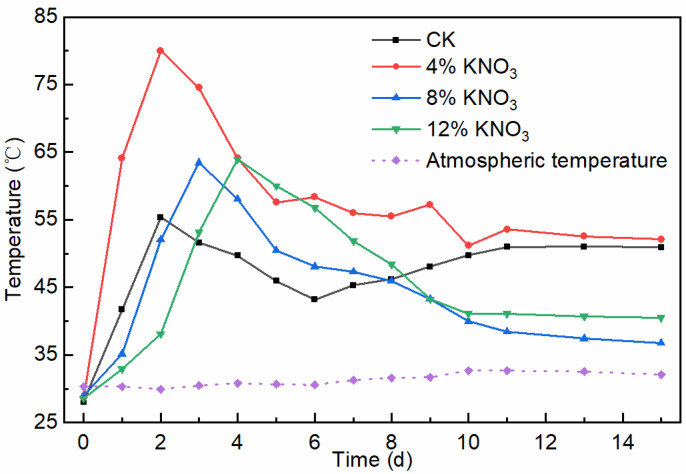
Changes in the composting pile temperature under different KNO_3_ additions.

**Figure 3 bioengineering-09-00258-f003:**
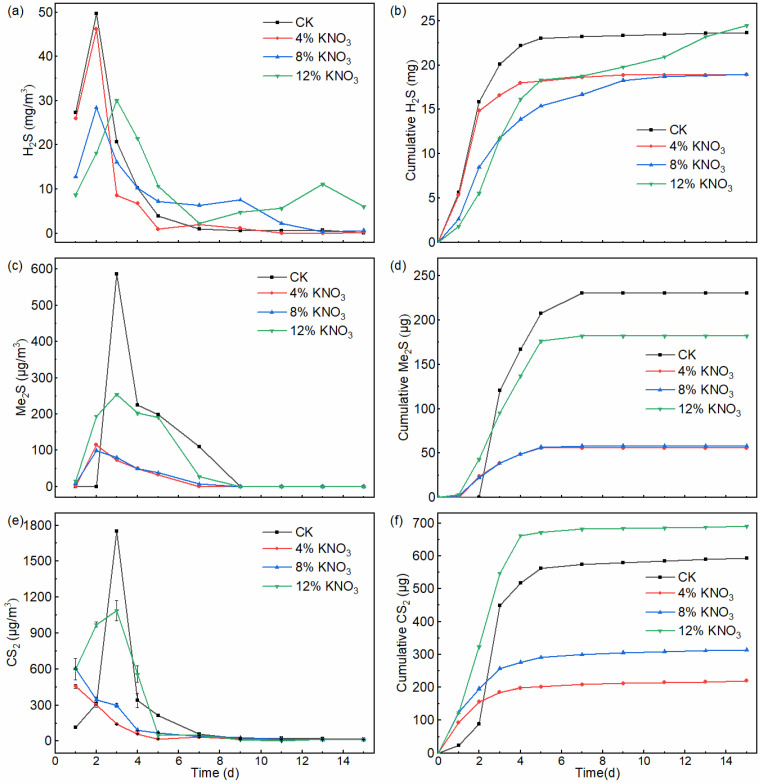
The dynamics of H_2_S (**a**), DMS (**b**), and CS_2_ (**c**) emissions and the cumulative emission of H_2_S (**d**), DMS (**e**), and CS_2_ (**f**) from the composting pile under different KNO_3_ additions.

**Figure 4 bioengineering-09-00258-f004:**
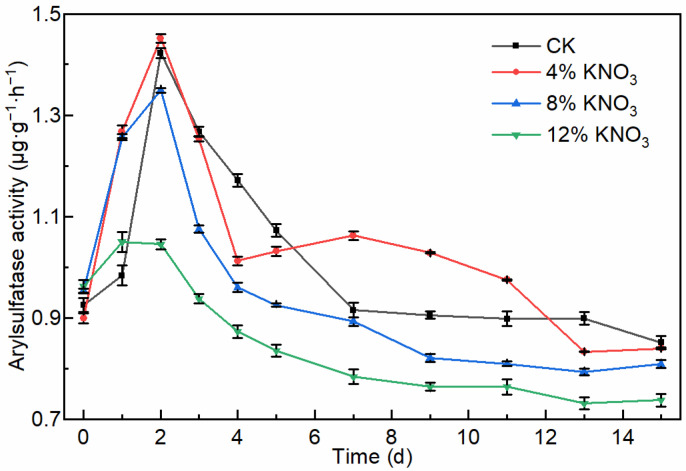
Variation in the arylsulfatase activity of the compost material under different KNO_3_ additions.

**Figure 5 bioengineering-09-00258-f005:**
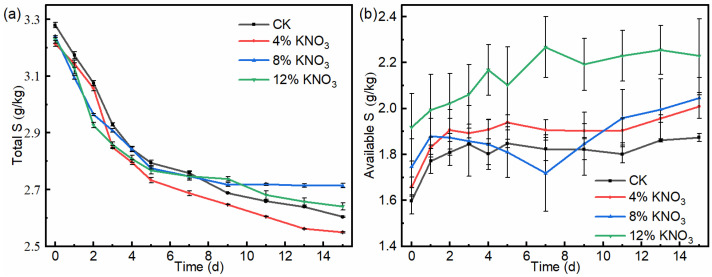
Variation of the total sulfur (**a**) and available sulfur (**b**) contents of the compost material under different KNO_3_ additions.

**Figure 6 bioengineering-09-00258-f006:**
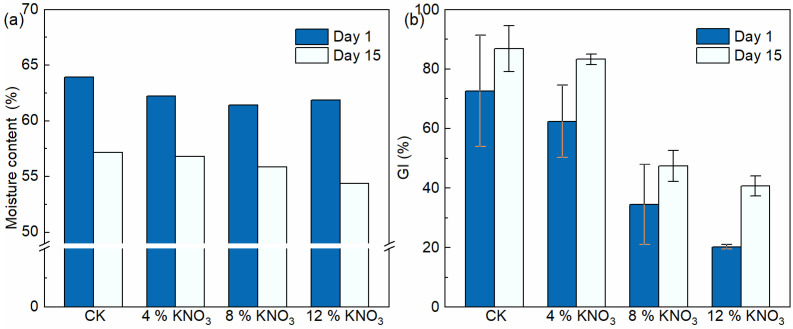
Changes in the moisture content (**a**) and GI (**b**) of the compost material under different KNO_3_ additions.

## Data Availability

Not applicable.

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
