# Peer review of "Inhibitory Effects of the Addition of KNO3 on Volatile Sulfur Compound Emissions during Sewage Sludge Composting"

_bioengineering, 2022, doi:10.3390/bioengineering9060258_

Round 1

Reviewer 1 Report

The study investigated the effect of adding KNO3 on the emissions of volatile sulfur compounds such as H2S, dimethyl sulfide, and carbon disulfide during sewage sludge composting, and on the physicochemical properties of compost products. The results showed that the addition of KNO3 could inhibit the emissions of volatile sulfur compounds during composting and adding KNO3 to compost can solve the problem of potassium deficiency when compost products are used as an organic fertilizer. This study is interesting and I think this work can be useful for other researchers after modifications.

Specific comments

Line 18 "thereby" is used inappropriately and without obvious causal logic

Line 21 Please apply the abbreviation with reasonable accuracy, for example "CS2"

Line 28 "anaerobic effect" is an unfamiliar expression.

Line 35 "large emissions", a formulation that may not be accurate.

Line 43 The number of significant digits finally matches the context.

Line 44 "a certain amount of ", inaccurate expression.

Line 83 GC-MS determination procedure suggested additional references or descriptions.

Line 90 GI determination procedure suggested additional references or descriptions.

Line 97 Suggest echoing, where appropriate, the hypothesis that Line 49 "the poor liquidity of the system may adversely affect the KNO3-inhibiting activity of SRB"

Line 98-100 Removal of irrelevant content

Line 106 Cross-reference error.

Line 116 Please amend "the addition of NO3-" to the uniform formulation above.

Line 117 The "respiration" may not be accurate.

Line 141 Please amend "treatments of the addition of 4%, 8%, and 12%" to the uniform formulation above.

Line 152-154 The second and third reasons will be further explained in additional detail.

Line 259 The number of significant digits finally matches the context.

Line 284 "H2S, Me2S, and CS2" may be removed.

Line 285 "It also promotes the decomposition and dewatering of the compost pile", exaggerating the previous conclusion.

Author Response

Dear editors and the reviewers,

We would like to express our appreciation to you and the reviewers for your suggestions on how to improve our manuscript titled “Inhibitory effects of the addition of KNO3 on volatile sulfur compound emissions during sewage sludge composting" (ID: bioengineering-1731965). Those comments were valuable and very helpful for revising and improving our paper, as well as providing important guiding significance for our study. We have studied the comments carefully and made corrections that we hope will meet with your approval. The revised portions are visible which marked in red in the paper. The main corrections to the paper and the responses to the editors and the reviewer’s comments appear below.

Reviewer 2 Report

The authors studied the effect of adding KNO3 on emissions of malodorous sulfur compounds.  I agree with the authors that the malodorous emissions from composting is the biggest thread its widespread utilization; any means to reduce such emissions at a reasonable cost will help the industry significantly.  However, I feel like the authors had designed their study poorly resulting significant shortcomings in findings.

My specific comments are below:

Abstract:

L 12: consider process rather than technology.

L 13-14, and throughout the manuscript: authors use chemical names, chemical formulas or common names all over.  I suggest sticking to one form to have uniformity.  Furthermore, authors use Me2S for dimethylsulfide—never seen this before.

L 16-18: Authors stipulate that increase on heating rate and peak temperatures are the cause of the sulfur loss.  I think this is way far over reaching considering they didn’t control other potential variables.  There could be several other reasons for reduction of S loss.

Introduction:

L28: Consider anaerobic pockets rather than effects.

L30: Consider odorous instead of odor in front of gases.

L36: Consider using odor “detection” thresholds.

L44: …certain amount of Fe2O3; instead of word “certain” authors should put the actual does.

L48: Consider “solid nature” instead of poor liquidity.

L53-53: Sentence doesn’t sound right.

L62-63: Consider “The findings of this study provides…..”

Materials and Methods

Generally speaking, the Materials and Method section doesn’t provide any useful information.  The point of this part is that anyone should be able replicate the study with very minimal search beyond the manuscript.  Authors talk about a GCMS but don’t even mention what GCMS, or its conditions; how the emission samples were collected; how they were introduced into the GCMS.  There is no information regarding the properties of the raw materials—at least, the C, N, S, moisture content should be provided along with the final products.  Also, no mixing for two weeks.  A lot is going on during the first two weeks of composting; in order to ensure uniformity, they should have been mixed. 

Perhaps a diagram of the reactors would be good to show here.  More importantly, I thing there are two major flaws in the experimental design.  First, it shouldn’t have been cut off at 15 days.  I agree that majority of the emissions occur during the initial 2 weeks of the process. However, since the study looks at the effect of additives, their effect on final product is important as is the emission reduction.

More importantly, the experiments should have been conducted in replications—this is particularly important considering the ununiform nature of the composting matrix.  Collecting emissions and analyzing them in analytical instrument in replication doesn’t provide much since the accuracy of analytical devices very high--the error or std.dev. will be very low.

L71: Revise the sentence.. Possibly: The composting experiments were conducted in automatically controlled reactors (or containers).

L75: I am guessing authors meant 1 hour not minute.

L76: Please explain how the materials were mixed

L78: Consider “batch or experiment” instead of group.

L78-81: This section should be rewritten as it is hard to read. 

L92: Authors look at the Germination Index (GI) as a sole measure of final product quality whereas there more and better ways to look at the final product.

Also, authors talk about the acrlysulfatase activity however never mention how it was measured. 

Results and “DISCUSSION” is missing

L 98-100: Don’t belong there, should be deleted.

L 104: In two places authors state “certain;”  considering this is a scientific manuscript, they should write more definitive. 

L 106: I assume authors mean Figure 1 rather than the bold text.

L 108: Consider “thermophilic” instead of high-temp.

I am concerned the very high-temperature in one of the runs.  Above 65C, the microbial community starting suffer—being killed/inactivated which could have affect the outcome.  Generally speaking, the temperatures are controlled below 65C by means of increased aeration or turning over.  Furthermore, the temperature readings were taken only once a day.  During the initial stage of composting, temperatures change substantially.  I would bet that authors probably miss the real peak temperature.  What was automated in the design?  Temperature readings should have been taken with a higher frequency.

Since there are several oversights throughout, I am concerned that the authors don’t understand the complexities of composting, thus didn’t do a good job designing their study.

L119-120:This statement is way too far reaching.  The higher temperatures/peak temp could be due occurrence of channeling, ununiform aeration and cooling. 

All of the figures should be redone.  Impossible to see which line is which!

Line 131: Consider “malodorous compound emissions” rather than odor emissions.

Line 133: Similar trend to what?  Each other?  Again, with once a day temp readings, I would refrain making any solid recommendations based on it.

Line 146: Delete “the” from “In the composting….”

Line 150: How much the reduction was, it should be stated.

Line 155: Author mention the “fourth reason.”  It is just a possibility; therefore should be mentioned as such.

Line158-160: Not clear, I suggest rewriting it.

Figure 2: Again, I don’t think the error bars represent the variation between the treatments but rather variation between the measurements of the same sample.  Therefore, I am concerned that whether the differences are statistically significant.

Line 164: I am not sure if authors chosed the correct word by “stimulate.”  I felt may be they meant “irritate.”  CS2 is also poison.

Line 176-177: Again I am concerned with how much emphasis is placed on temperatures considering that bulk of the temp profile may not be representative.

Line 181.  Consider “batch” rather than group.

Line 191: Consider “metabolism” or “pathway” rather than mechanism.

As I am reading the manuscript, I cannot help but think that 8-12% addition by weight of anything is a massive amount and a major cost.  Is it even feasible to add so much.  How these rates were chosen, based on what?  Furthermore authors didn’t do a through work on the effect of KNO3 on microbial activity, pathways particularly, the effect on nitrogen pathways and nitrogenous compound emissions.  Also, as suggested by the authors, adding so much of an ionizing compound could have had adverse effect the microbial activity due to osmotic pressure caused by ionic strength.  These should have been looked at. 

Line 200 and around, authors talk about, and compare, the emissions and reductions of H2S and Me2S.  Comparing the percentages could be, and is, misleading.  If they were to look at the actual amounts, and perform a mass balance analysis, they could have seen better.  Furthermore, during the composting, the material loss is significant; so just looking at and comparing before and after percentages could be misleading.

Line 242:  What is effective sulfur mean? Available?  And how it was measured/determined.

Line 253.  Authors stated that the moisture content directly determines the speed of the aerobic composting process.  Yes, the moisture content has a lot to do with the rate of microbial activity.  However, it doesn’t determine “directly.”  They didn’t even determine moisture content during the experiment but only looked at the before and after 15 days.  It is also hard to visualize that only 10% or so moisture is lost after aerating, and at such a high temperatures.    

Line 273-276: Authors contradict themselves.  They are suggesting that the presence of large amounts of KNO3 could have inhibited the microbial activity (again it is only measured by temp); then how come addition of 8% didn’t do that?

Line 283: Instead of saying “appropriate” authors should have given an amount.  Appropriate is way too loose term.

Line 285-286: Author’s claim is just a speculation; not based on any data. 

Author Response

(The authors gave the same response as above.)
